# Three-Dimensional High-Precision Numerical Simulations of Free-Product DNAPL Extraction in Potential Emergency Scenarios: A Test Study in a PCE-Contaminated Alluvial Aquifer (Parma, Northern Italy)

**Alessandra Feo** , **Riccardo Pinardi \***, **Andrea Artoni** **and Fulvio Celico**

Department of Chemistry, Life Sciences and Environmental Sustainability, University of Parma,
43124 Parma, Italy; alessandra.feo@unipr.it (A.F.); andrea.artoni@unipr.it (A.A.); fulvio.celico@unipr.it (F.C.)
**\*** Correspondence: riccardo.pinardi@studenti.unipr.it

**Abstract:** Chlorinated organic compounds are widespread aquifer contaminants. They are known to be dense non-aqueous phase liquids (DNAPLs). Therefore, they are denser than water and immiscible with other fluids. Their migration into the environment in variably saturated zones can cause severe damage. For this reason, optimizing those actions that minimize the negative impact of these compounds in the subsurface is essential. This paper presented a numerical model simulating the free-product DNAPL migration and extraction through a purpose-designed pumping well in a potential emergency scenario. The numerical simulations were performed using CactusHydro, a numerical code that uses a high-resolution shock-capturing flux conservative method to resolve the non-linear coupled partial differential equations of a three-phase immiscible fluid flow recently proposed in the literature, including the contaminant extraction at the base of the aquifer. We investigated the temporal (and spatial) evolution of its migration in the Parma (Northern Italy) porous alluvial aquifer following the saturation contour profiles of the three-phase fluid flow in variably saturated zones. The results indicated that this numerical approach can simulate the contaminant migration in the subsurface and the pumping of the free-product from a well screened at the base of the aquifer system. Moreover, the simulation showed the possibility of recovering about two-thirds of the free-product, in agreement with the scientific literature.

**Keywords:** DNAPL migration; DNAPL extraction; numerical simulations; groundwater immiscible flow; alluvial aquifer

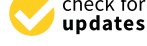



## 1. Introduction

Chlorinated organic compounds are widespread aquifer contaminants. They are referred to as dense non-aqueous phase liquids (DNAPLs); therefore, they are denser than water, with very low water solubility, capable of migrating under pressure and gravity forces through both the unsaturated and the saturated aquifer systems until reaching a bottom aquiclude (e.g., [1]). They are usually detected in industrial and urban areas due to their extensive application in chemical production, dry cleaning, and metal degreasing. Several of these compounds are of much concern as records of the Stockholm Convention on Persistent Organic Pollutants (POP). They are persistent in the environment and are linked to toxic and sometimes carcinogenic effects (e.g., [2]). Epidemiological studies were widely carried out from 1960 onward in Europe and the United States (e.g., in the dry-cleaning industry) to evaluate how exposure to chlorinated compounds can be associated with cancers. Different studies provided epidemiological evidence of the association between this type of exposure and bladder cancer, non-Hodgkin lymphoma, and multiple myeloma (e.g., [3]). At the same time, experimental evidence of carcinogenicity was evaluated using rats as reference organisms. For example, a significant increase in the incidence

of mononuclear cell leukemia was observed by Thomas et al. (2007) [4]. An increase in kidney tumors, usually rare in rats, was demonstrated by Lash and Parker (2001) [5] using tetrachloroethylene. Another rare rat cancer, brain glioma, was also increased, as well as the testicular interstitial cell tumors in male rats [3].

The behavior and fate of chlorinated organic compounds in the subsurface have been studied since the early 1980s (e.g., [6–11]). Some numerical models, including migration [12] in fractured aquifers [13,14] and remediation of alluvial aquifers [15], using the MT3DMS numerical code [16,17], have been written to simulate their migration in aquifer systems.

The dynamics of a spilled DNAPL migration in a variably saturated zone can be described using numerical simulations for the governing equations of immiscible phase fluid flow in a porous medium. These are coupled with conserved partial differential equations for each fluid flow, based on the Darcy equation together with the conservation of mass and an equation of state. They are written as a function of each fluid flow's saturation, capillary pressure, density, viscosity, permeability, and porosity.

Since the capillary pressure and permeability of each phase are a function of the saturation, these equations are non-linear with a dominant hyperbolic advection term proportional to gravity and the pressure gradient. It is responsible for forming sharp (shocks) front and rarefaction, which can create significant errors in the output results if not treated with conservative numerical solutions methods. See, for example, recent developments in transport modeling in a porous medium, including the presence of NAPL [18,19], immiscible fluid flow in unsaturated zones [20], using a high-resolution central upwind scheme for two-phase fluid flow [21], using a finite volume WENO scheme and discontinuous Galerkin methods [22], and a second-order accurate difference method for non-linear conservation laws [23].

This study deals with the 3D numerical model implemented to optimize the free-product DNAPL pool extraction from a shallow and unconfined alluvial aquifer using the method introduced in Feo and Celico [24,25] and validated through a laboratory experiment [26]. It is based on the high-resolution shock-capturing flux (HRSC) conservative method [27–29] to follow sharp discontinuities accurately and temporal dynamics of a three-phase immiscible fluid flow in a porous medium. CactusHydro is based on the Cactus computational toolkit [30–32], an open-source software framework for developing high-performance computing (HPC) simulation codes, and the data are evolved on a cartesian mesh using Carpet [33,34]. Several validation tests were performed to verify the accuracy of the HRSC method and the CactusHydro code.

The model presented here simulated the free-product DNAPL extraction through a purpose-designed pumping well in a potential emergency scenario to minimize the negative impact on the surrounding environment caused by a DNAPL release. The test site was the southern end of the Functional Urban Area of Parma (the city plus its commuting zone, *sensu* EU-Organisation for Economic Co-operation and Development), where perchloroethylene (PCE) was recently detected in the shallow groundwater [35,36], therefore suggesting the existence of DNAPL sources.

Cherry et al. [37] suggested that DNAPL pools are sometimes large enough to effectively pump the free-product from wells screened at the bottom of the pool. At the same time, free-product pumping usually recovers up to two-thirds of the DNAPL, leaving abundant DNAPL acting as a long-term pollution source. This is why the free-product extraction has been designed within a broader interdisciplinary approach that will also consider the natural attenuation and (briefly) bioremediation techniques.

## 2. Materials and Methods

### 2.1. Study Area

The research is carried out in the alluvial aquifer developing from the northern Apennine topographic margin to the Parma plain and laying above the Emilian Folds and the Salsomaggiore thrust-related anticline (Northern Italy; Figure 1). The northern Apennines are a fold-and-thrust belt composed of a pile of NE-verging tectonic units that developed

due to the Cenozoic collision between the European plate (Corso–Sardinian block) and the Adria plate. The tectonic units belong to the Ligurian, Tuscan, and Umbria-Romagna domains. The Ligurian units represent the uppermost tectonic units in the Apennine nappe pile and correspond to allochthonous terrains initially deposited in an oceanic realm (the Ligurian–Piedmontese sector of the Alpine Tethyan Ocean) composed of ophiolites and their Jurassic to Eocene sedimentary cover. These units tectonically overlie the Tuscan and Umbria–Romagna units, which, deposited initially on the passive margin of the Adria plate from the middle Triassic to early Cretaceous and convergent to the collisional margin from the middle Cretaceous to the present, consist of a lower succession of carbonate rocks from the Mesozoic–Cenozoic age and a thick upper succession of siliciclastic foredeep sediments from the Oligocene–Miocene age [38,39].

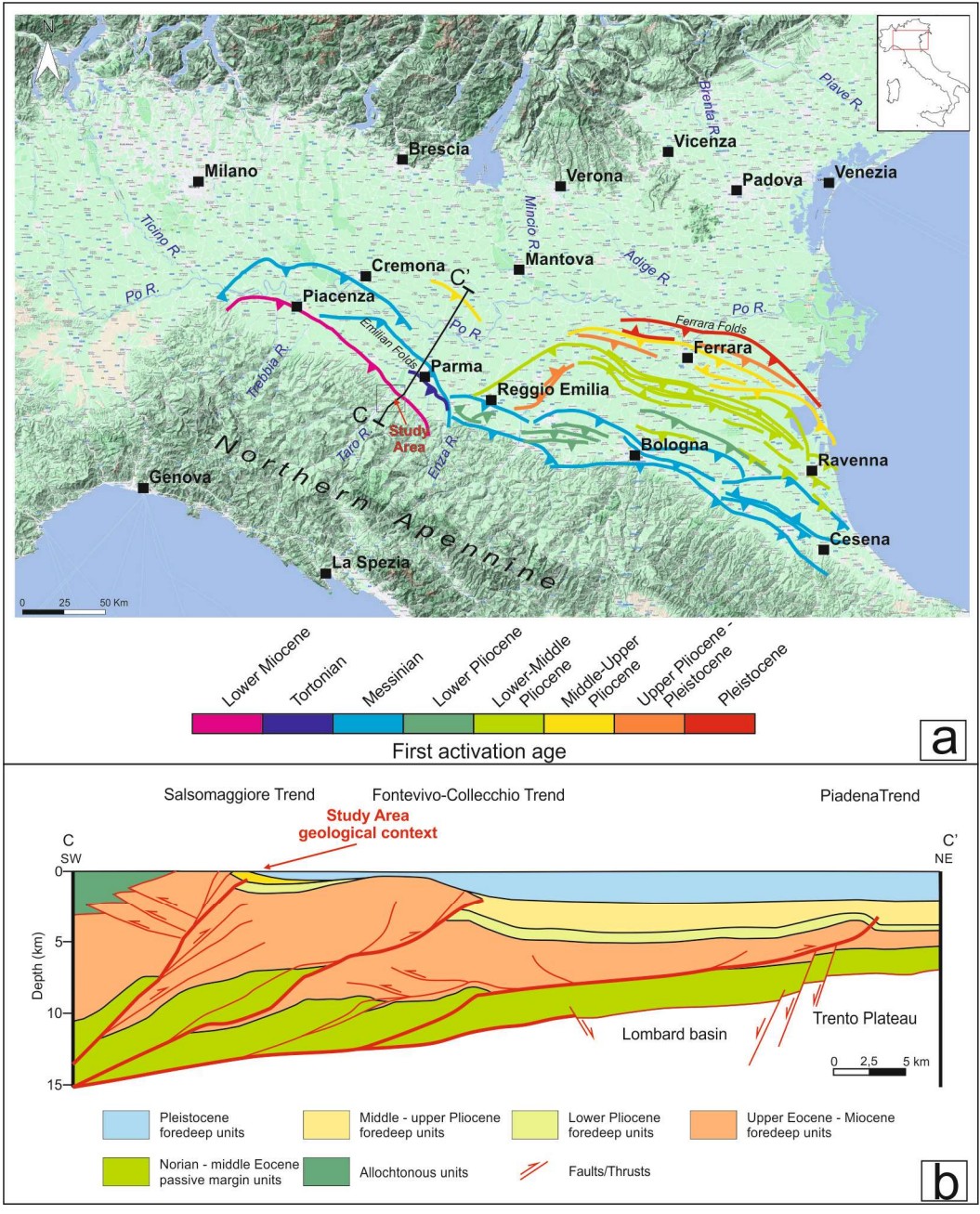

**Figure 1.** Geological-structural setting modified from [38,39]. (**a**) Location of major buried thrusts front near the study area. (**b**) Representative geological section.

During the orogenetic uplift from the Eocene to the Messinian, episutural and wedge-top basins were set on top of the Ligurian units, giving rise to the Epiligurian succession.

At the front of the chain, from the Messinian to the present, the Po Basin represents the northern Apennines foredeep, filled by the Plio–Pleistocene turbidite and deltaic to the alluvial syntectonic marine-regressive sequences, heavily influenced by the uplifts of the several thrust fronts buried under the plain [38–42].

The study area, along with the upper Emilia–Romagna plain and Apennine foothills, is mostly characterized by this marine-to-continental regressive sedimentary succession. The sequence of units (or synthems *sensu* [43]) is characterized at the base by the hectometric thicknesses of Pliocene clay marine sediments (FAA), which are discordantly overlapped by progressively more continental deposits, from shallow marine and fan-deltas (ATS and CMZ) to alluvial plain and foothill alluvial fans (AEI and AES), during the Pleistocene until the present (Figure 2).

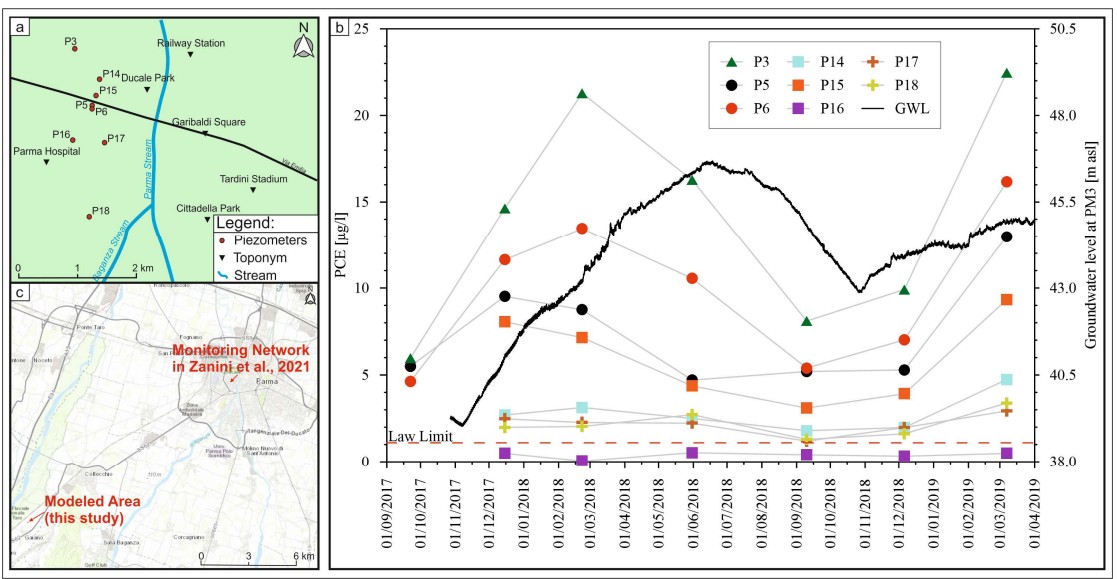

**Figure 2.** PCE concentrations in the shallow groundwater ((**a**) location of observation wells; (**b**) PCE concentrations vs. time (data are given in day/month/year); (**c**) location of the modeled area in the present study and the monitoring network in Zanini et al., 2021; from [36], modified).

The studied aquifer corresponds to the Pleistocene alluvial synthem, lithologically defined by alternating gravels, sands, silts, and silty clays, and generated in the area by the depositional dynamics of the ancient Taro River. At the bottom of the succession, Pliocene marine clays' high thickness is considered an extensive regional-scale aquiclude [44].

The contact surface between the aquifer and basal aquiclude, approximately corresponding to the Pliocene–Pleistocene boundary, is characterized by several undulations due to the Apennine tectonic compressive action and subsequent erosion of the Emilian Folds.

Within the studied aquifer system, the shallow groundwater flows from southwest to northeast at a basin scale [35] and is recharged by (i) local effective infiltration and (ii) the Taro River, which is hydraulically connected to the groundwater [35,36,45–47]. The interconnection between the river and groundwater is also supported through analyses of stable isotopes ($\delta^{18}O$ and $\delta^2H$; [48]. Some well tests allowed us to estimate the studied aquifer's hydraulic conductivity, which ranges between $1.2 \times 10^{-5}$ and $4.9 \times 10^{-5}$ m/s; [35,36].

Concerning groundwater contamination, PCE was detected from 2002 to 2019, with concentrations up to 23 µg/L [35,36], using institutional monitoring wells (Parma Municipality), integrated with new ones within the AMIIGA CE32 research project (UE INTERREG program). During the last years, frequent monitoring of PCE concentrations at several wells showed a cyclic variation, with the highest values at the beginning of recharge and the lowest ones in late recession (Figure 2). The persistence of PCE in the groundwater for over

two decades at least, the cyclic variation of PCE concentrations over time as observed from 2017 to 2019, and the nearly-constant peak concentration observed in different hydrologic years suggest the existence of free-product DNAPL pools' upgradient regarding the wells used by Zanini et al., 2001 [36].

Therefore, the 3D high-precision simulation of DNAPL migration has been performed in the southern end of the shallow alluvial aquifer to design the best set up of the hydraulic barriers to be constructed in emergency scenarios and/or to extract existing pools once identified based on the spatial distribution of PCE concentrations in groundwater and a detailed reconstruction of the base aquiclude morphology.

### 2.2. Geological and Hydrogeological Data

Both the geological map (Figure 3) and the sections reconstructed during the present research are based on a re-interpretation (from a hydrogeological perspective) of the stratigraphic profiles derived from the published database by Regione Emilia–Romagna.

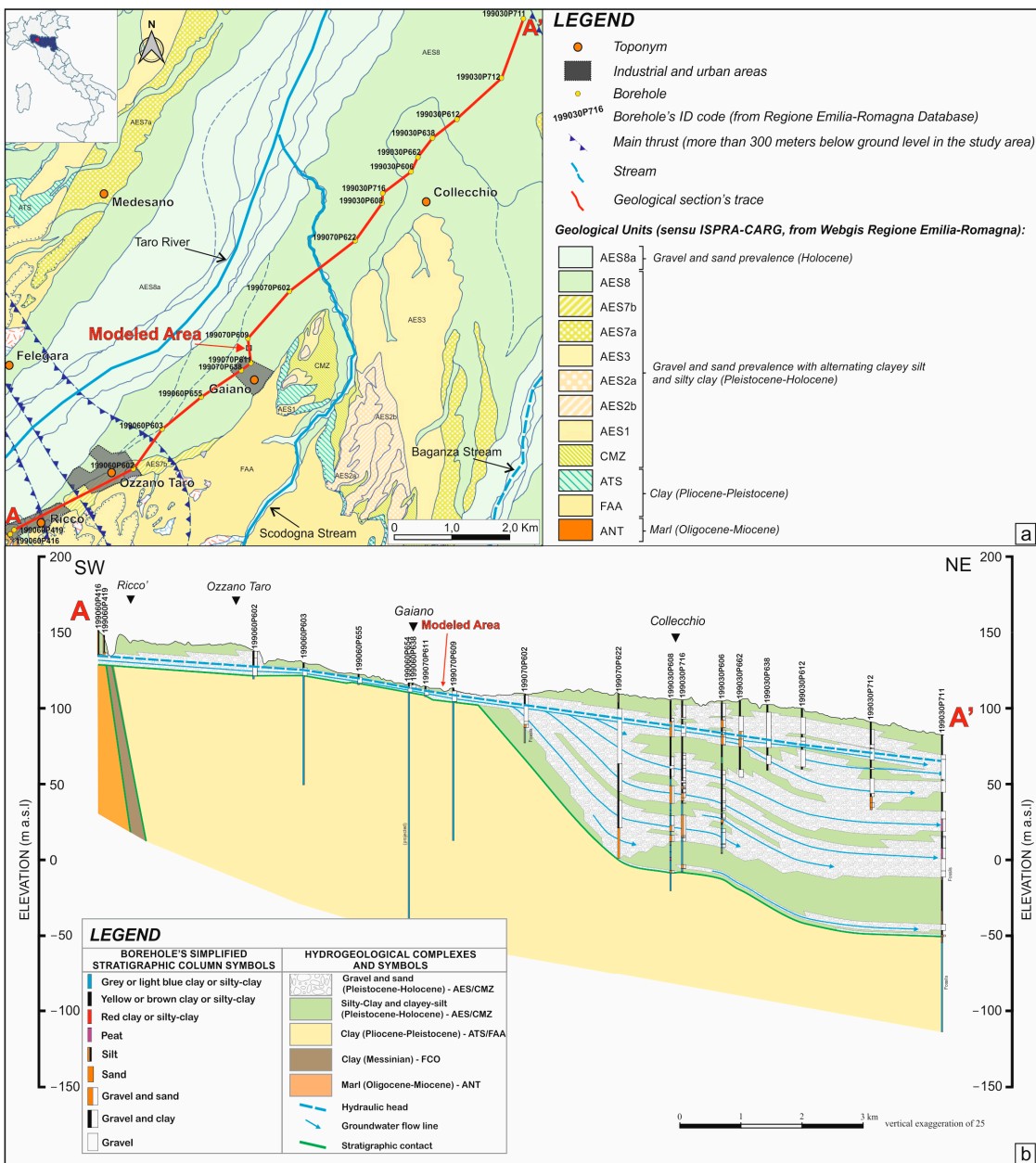

**Figure 3.** (**a**) Geological map with section trace (AA'). (**b**) Geological/hydrogeological section AA'.

At first, after an accurate selection of the subsurface as mentioned in the above data, a stratigraphic section is constructed, having an orthogonal direction to the Apennine front (SW-NE), to obtain a self-constructed perspective of the geological (and hydrogeological) context at the regional scale, in which the investigated aquifer is located. Then, thanks to the regional section, it is possible to identify (and characterize in more detail) the area from which possible DNAPL releases could cause a widespread and deep groundwater contamination downgradient. Within this study process, the aquifer bottom morphology has been a nodal issue, considering the ability of DNAPLs to migrate rapidly above the aquicludes.

Therefore, starting from four detailed stratigraphic sections, a 3D geological reconstruction of this contact surface is carried out. The aquifer bottom surface resulting from the 3D geological model is transferred to the numerical model.

### 2.3. Governing Equations and Mathematical Setup

The governing equations that describe a multiphase fluid flow in a porous medium in terms of non-aqueous (*n*), water (*w*), and air (*a*) were introduced in Refs. [24,25] and are given by,

$$\frac{\partial}{\partial t}(\rho_n \phi S_n) = \frac{\partial}{\partial x^i}\left[\rho_n \frac{k_{rn}}{\mu_n}k^{ij}\left(\frac{\partial p_a}{\partial x^j} + \rho_n g\frac{\partial z}{\partial x^j}\right)\right] - \frac{\partial}{\partial x^i}\left[\rho_n \frac{k_{rn}}{\mu_n}k^{ij}\left(\frac{\partial p_{can}}{\partial x^j}\right)\right] + q_n, \tag{1}$$

$$\frac{\partial}{\partial t}(\rho_w \phi S_w) = \frac{\partial}{\partial x^i}\left[\rho_w \frac{k_{rw}}{\mu_w}k^{ij}\left(\frac{\partial p_a}{\partial x^j} + \rho_w g\frac{\partial z}{\partial x^j}\right)\right] - \frac{\partial}{\partial x^i}\left[\rho_w \frac{k_{rw}}{\mu_w}k^{ij}\left(\frac{\partial p_{caw}}{\partial x^j}\right)\right] + q_w, \tag{2}$$

$$\frac{\partial}{\partial t}(\rho_a \phi S_a) = \frac{\partial}{\partial x^i}\left[\rho_a \frac{k_{ra}}{\mu_a}k^{ij}\left(\frac{\partial p_a}{\partial x^j} + \rho_a g\frac{\partial z}{\partial x^j}\right)\right] + q_a, \tag{3}$$

where $x^i = (x, y, z)$ are the spatial cartesian coordinates, and $t$ is the time coordinate. $\rho_n$, $\rho_w$, $\rho_a$, $\left[\frac{M}{L^3}\right]$, are the density of the non-aqueous, water, and air, respectively. $\mu_n$, $\mu_w$, $\mu_a$, $\left[\frac{M}{LT}\right]$, are the dynamic viscosity of the non-aqueous, water, and air, respectively. $k^{ij}$ $\left[L^2\right]$, $g$ $\left[\frac{L}{T^2}\right]$, $z$ $[L]$, $q_\alpha$ $\left[\frac{M}{L}\right]$, $\phi$, are the absolute permeability tensor, the gravity acceleration, the depth, the mass source/sink, and the porosity, respectively. A fourth equation relates the saturation of the different phases and is given by

$$S_w + S_n + S_a = 1. \tag{4}$$

The system (1)–(4) is written in terms of the pressure of the air, $p_a\left[\frac{M}{T^2 L}\right]$; the saturations; the capillary pressure for the air–non-aqueous phase, $p_{can} = (p_a - p_n)$; and the capillary pressure for the air–water phase, $p_{caw} = (p_a - p_w)$. The relative permeabilities $k_{rw}$, $k_{rn}$, and $k_{ra}$, and the capillary pressures are a function of the saturations, $k_{r\alpha} = k_{r\alpha}(S_a, S_n, S_w)$, $p_{can} = p_{can}(S_a, S_n, S_w)$, and $p_{caw} = p_{caw}(S_a, S_n, S_w)$. See the following subsection.

The porosity $\phi$ is a function of pressure and can be linearly approximated to $\phi = \phi_0[1 + c_R(p - p_0)]$, where $c_R$ is the rock compressibility, $\phi_0$ is the porosity at $p_0$, which is considered the atmospheric, and $p$ is the pressure (that will be associated with $p_a$). Let us define the product of the porosity $\phi$ and the saturation for each phase as $\sigma_w \equiv \phi S_w$, $\sigma_n \equiv \phi S_n$, $\sigma_a \equiv \phi S_a$. Equation (4) can be written as $\sigma_a + \sigma_n + \sigma_w = \phi_0[1 + c_R(p - p_0)]$. The left-hand side of (1)–(3), assuming a constant density-viscosity for each phase, becomes

$$\frac{\partial \sigma_a}{\partial t} + \frac{\partial \sigma_n}{\partial t} + \frac{\partial \sigma_w}{\partial t} = \phi_0 c_R \frac{\partial p}{\partial t}, \tag{5}$$

with

$$\frac{\partial \sigma_{(\alpha)}}{\partial t} = -\frac{\partial}{\partial x^i}\left[F^i_{(\alpha)}(S_w, S_n, S_a, p)\right] + \frac{\partial}{\partial x^i}\left[Q^i_{(\alpha)}(S_w, S_n, S_a, p)\right] \tag{6}$$

and $\alpha = (w, n, a)$:

$$F^i_{(\alpha)}(S_w, S_n, S_a, p) = -\frac{k_{r(\alpha)}(S_w, S_n, S_a)}{\mu_{(\alpha)}}k^{ij}\left(\frac{\partial p}{\partial x^j} + \rho_\alpha g\frac{\partial z}{\partial x^j}\right), \tag{7}$$

does not depend on the spatial derivative of the saturation, and

$$Q^i_{(\alpha)}(S_w, S_n, S_a, p) = -\frac{k_{r(\alpha)}(S_w, S_n, S_a)}{\mu_{(\alpha)}}k^{ij}\frac{\partial p_{ca(\alpha)}(S_w, S_n, S_a)}{\partial x^j}, \tag{8}$$

depends on the spatial derivative of the saturation. The partial differential equation system to be numerically resolved is then (5) and (6) and variables $p$, $\sigma_w$, $\sigma_n$, $\sigma_a$ together with the non-linear functional form of the relative permeabilities and capillary pressures.

We separated the right-hand side of the system (6) in the advection (hyperbolic) partial differential equation, the one that depends on the derivative of the pressure and the gravity (Equation (7)), and the parabolic partial differential equation, the one proportional to the capillary pressures (Equation (8)) in the variable saturation. These two pieces are treated differently when we use numerical methods. The hyperbolic part is responsible for the shock formation when the flow passes through a discontinuity and must be treated using a mass-conservative numerical method.

This separation is important since explicit methods in the time evolution should be preferred when propagation dominates over the diffusion. The importance of using conservative formulations, i.e., methods based on conservation law, is due to two fundamental theorems in Refs. [28,29]. These two theorems state that if a conservative formulation is used, we are guaranteed that the numerical solution will converge to the correct one. In contrast, if a conservative formulation is not used, we are guaranteed to converge to the incorrect solution if the flow develops a discontinuity.

The method used is HRSC flux conservative and belongs to the class of the Monotonic Upstream-Centered Scheme for Conservation Law (MUSCL) suggested by van Leer in 1973, and the Kurganov–Tadmor scheme [27] that is up-to-second-order accurate in space using analytical information on the form of the flux. However, as in this work, using the first-order upwind formula for the fluxes, and the minmod flux limiter, only the point values of the flux are required. This is of great advantage since it can use tabulated values for permeabilities. Consequently, the integration in time is performed in the first-order (for consistency) using the Euler method.

A model simplification is that at this stage, we consider these fluids immiscible, which means that volatilization and/or dissolution processes are not considered.

### 2.4. Hydrogeological Parameters of the Simulation Model

A 3D numerical model is set up using the numerical code CactusHydro, introduced in [24,25]. CactusHydro resolves the governing equations that describe the migration of an immiscible phase fluid flow in a porous medium composed of non-aqueous ($n$), water ($w$), and air ($a$), and a variably saturated zone. The migration of the spilled DNAPL is considered immiscible, and the effects of the volatilization, biodegradation, or dissolution are not considered. CactusHydro treats the vertical and horizontal movement of the contaminant in the variably saturated zone as coupled and is numerically resolved as a unique zone (not separating the vertical movement from the horizontal one since the flow equation includes both zones).

Table 1 shows the DNAPL (PCE) phase properties such as density, viscosity, porosity, and parameter details used in the numerical simulations. In particular, the density of the DNAPL contaminant is $\rho_n = 1643\,\text{kg/m}^3$, and thus denser than water.

**Table 1.** Definitions of the parameters used in the numerical simulations of a DNAPL (PCE) migration in a variably saturated zone.

| Parameter | Symbol | Value |
|---|---|---|
| Absolute permeability | $k$ | $5.102 \times 10^{-12}$ m$^2$ |
| Rock compressibility | $c_R$ | $4.35 \times 10^{-7}$ Pa$^{-1}$ |
| Porosity | $\phi_0$ | 0.37 |
| Water viscosity | $\mu_w$ | $10^{-3}$ kg/(ms) |
| Water density | $\rho_w$ | $10^3$ kg/m$^3$ |
| DNAPL viscosity | $\mu_n$ | $0.844 \times 10^{-3}$ kg/(ms) |
| DNAPL density | $\rho_n$ | 1643 kg/m$^3$ |
| Air viscosity | $\mu_a$ | $1.8 \times 10^{-5}$ kg/(ms) |
| Air density | $\rho_a$ | 1.225 kg/m$^3$ |
| Van Genuchten | $(n, m)$ | $\left(2.68, 1 - \frac{1}{2.68}\right)$ |
| Irreducible wetting phase saturation | $S_{wir}$ | 0.045 |
| Surface tension air–water | $\sigma_{aw}$ | $7.199 \times 10^{-2}$ N/m |
| Interfacial tension non-aqueous–water | $\sigma_{nw}$ | $4.44 \times 10^{-2}$ N/m |
| Capillary pressure air–water at zero saturation | $p_{caw0}$ | 676.55 Pa |
| Capillary pressure air–non-aqueous at zero saturation | $p_{can0}$ | 259.83 Pa |

We consider a hydraulic conductivity value in the variably saturated zone equal to $K = 5.0 \times 10^{-5}$ m/s. From here, we determine the value of the absolute permeability in Table 1, using the relation $k = \frac{K \mu_w}{\rho_w g}$, all over the grid. For the impermeable zone, we consider a value of $k = 5.102 \times 10^{-19}$ m$^2$ [49].

For a three-phase fluid flow, we need two capillary pressures. In our formulation [22,23], we use the air–water phase, $p_{caw} = (p_a - p_w)$, and the capillary pressure for the air–non-aqueous phase $p_{can} = (p_a - p_n)$. Then, the non-aqueous–water capillary pressure is given by $p_{cnw} = (p_n - p_w) = (p_{caw} - p_{can})$. The relative permeabilities $k_{rw}$, $k_{rn}$, and $k_{ra}$, and the capillary pressures are a function of the saturations, $k_{r\alpha} = k_{r\alpha}(S_a, S_n, S_w)$, $p_{can} = p_{can}(S_a, S_n, S_w)$, and $p_{caw} = p_{caw}(S_a, S_n, S_w)$. For three phases, they have been extended from the two-phase expressions [8] (see [24,25]). For the capillary pressure, instead, we use the van Genuchten model [50] (see [24,25]).

The van Genuchten parameter is $\alpha = \left(\frac{p_c}{\rho_w g}\right)^{-1}$, where $p_c$ is the capillary pressure head. For "sand", the value is $\alpha = 0.145$ cm$^{-1} = 14.5$ m$^{-1}$, and the capillary pressure between air and water, at zero saturation, gives: $p_{caw} = \frac{\rho_w g}{\alpha} = \frac{10^3 \text{ kg/m}^3 \ 9.81 \text{ m/s}^2}{14.5 \text{ m}^{-1}} = 676.55$ Pa.

Using the value of the interfacial tension $\sigma_{nw} = 4.44 \times 10^{-2}$ N/m and $\sigma_{aw} = 7.199 \times 10^{-2}$ N/m we have that $\beta_{nw} = \frac{\sigma_{aw}}{\sigma_{nw}} = 1.62$, and $p_{cnw}(S_w) = \frac{p_{caw}}{\beta_{nw}} = 417.62$ Pa. Then, the capillary pressure at zero saturation is $p_{can} = p_{caw} - p_{cnw} = 259.83$ Pa.

## 3. Results

### 3.1. Geological and Hydrogeological Model

The geological model obtained through the re-interpretation of existing strati-graphic data agrees with the existing stratigraphic models [46] but allows refining the hydrogeological model at the local and broader scale to obtain the best high-precision numerical simulations as possible at the site scale and predict the potential impact of PCE release in a wider hydrodynamic scenario.

These reconstructions highlight the known marine-to-continental regressive sedimentary succession of the upper Parma plain and Apennine foothills.

Two- and three-dimensional geological reconstructions allow the recognition of the units and synthems described in the literature (FAA, ATS, CMZ, and AES in Figures 3 and 4). From a stratigraphic point of view, in the investigated area, the typical marine-regressive succession is recognized. Starting from the base, the presence of the hectometric thicknesses of marine blue clays with fossils is evident (see 199060P654 well-log, FAA in Figures 3 and 4), and fan-delta and foreshore facies are superimposed, evidenced by sands-

gravels and clayey silt cycles (see 199070P602 and 199070P622 well-logs basal parts, Figure 3). The transition from backshore lagoon, probably highlighted by the presence of green clayey silt typical of anoxic deposition (see traces in 199030P608 and 199030P606 well-logs, Figure 3), to facies typical of floodplains and mountain conoids with alternating gravels and silts-clays, is also recorded.

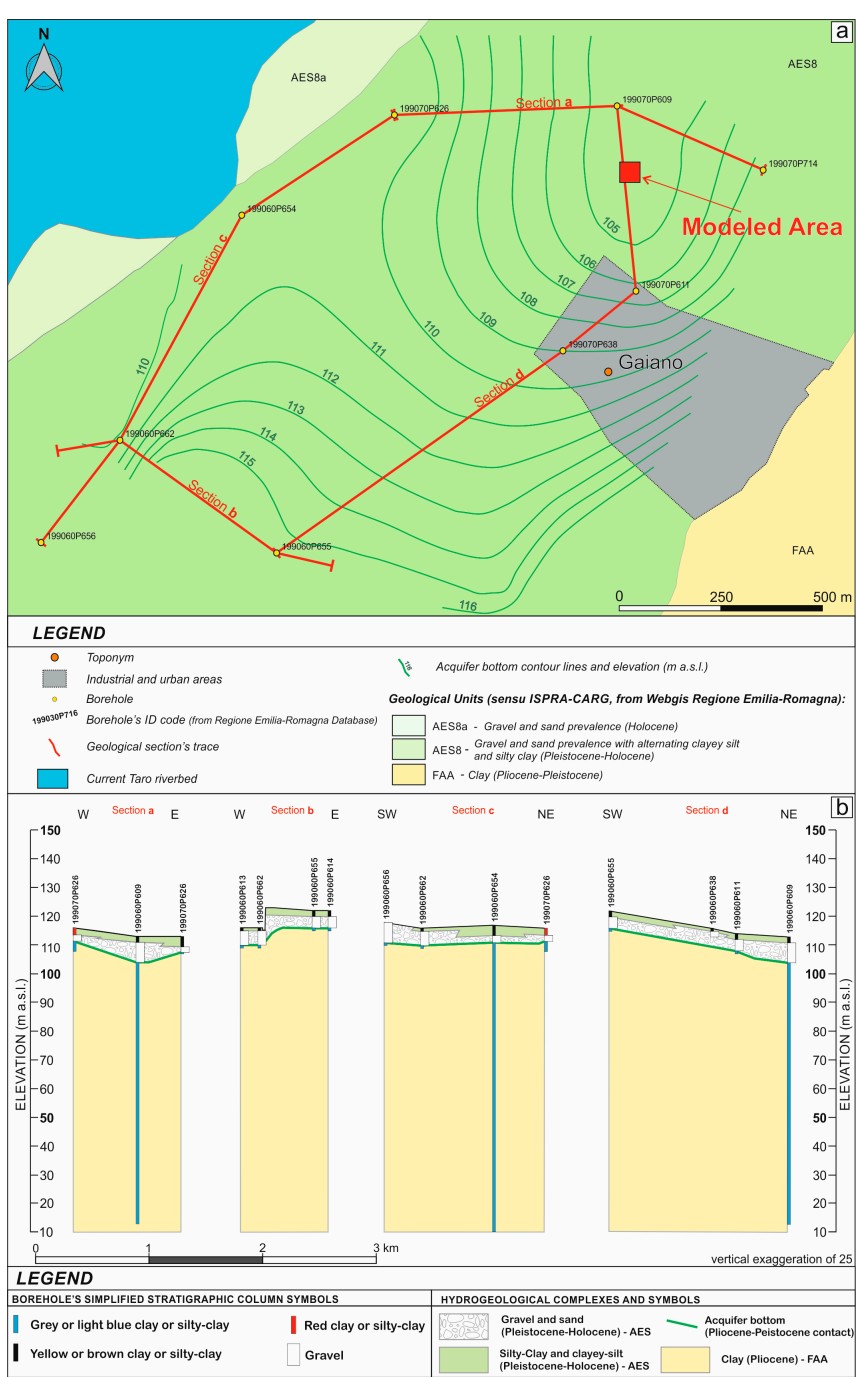

**Figure 4.** (**a**) Aquiclude base contour map and location of the modeled area. (**b**) Detailed stratigraphic sections (a, b, c, d).

From a structural point of view, the well-known arrangement of these aquifer units is also confirmed. In the southern zone of the investigated area, the alluvial terrace of the Taro River (AES8, the most recent unit in the area) is settled with a stratification parallel to the topographic plane. Proceeding towards the valley (to the north), between the villages

of Gaiano and Collecchio, a fan of gravel-sand-silt strata opens in-depth, with gradually smaller angles of inclination as the more recent depositions originated it. The resulting geometric and physical features of the aquifer arrangement validate its syntectonic nature, contemporary with the Apenninic tectonics' activations.

The analysis of the morphology of the aquifer base (Pliocene–Pleistocene boundary), starting from 3D geological modeling, leads to the construction of a contour map that highlights a depression in the base aquiclude (north of Gaiano village; Figure 4) where DNAPLs (potentially released within urban and industrial areas) would migrate downgradient. This area is of fundamental importance on a regional scale because it is located just upstream and in direct hydraulic connection with the whole heterogeneous aquifer to the south. Therefore, the same area is selected as the best test site to simulate a PCE release in the subsurface and the following free-product PCE pumping to minimize the negative impact on the surrounding environment.

### 3.2. Three-Dimensional Numerical Simulations Results

The 3D numerical model developed in this study is based on the conceptual model discussed in the previous subsection. It described the migration of a free-product PCE and its extraction from a shallow and unconfined alluvial aquifer using an HRSC conservative flux method and the CactusHydro code introduced in [24,25] with the addition of a module that allows the contaminant extraction over a fixed time interval. It is based on the reconstruction through interpolating the points of three surfaces: the aquifer bottom surface (clay-gravel contacts, below there is an aquiclude) and the upper-level surface (ground surface). A third surface between the previous surfaces is the groundwater table separating the unsaturated zone from the saturated aquifer. See Figure 5, which shows the three surfaces, where (a) the red one is the ground surface, and the blue one is the groundwater table surface. The green one is the aquifer bottom surface, and (b) the zoom of the aquifer bottom surface, which highlights a depression in the base aquiclude (see Figure 4) using the georeferenced coordinates and a grid spacing of 0.1 m.

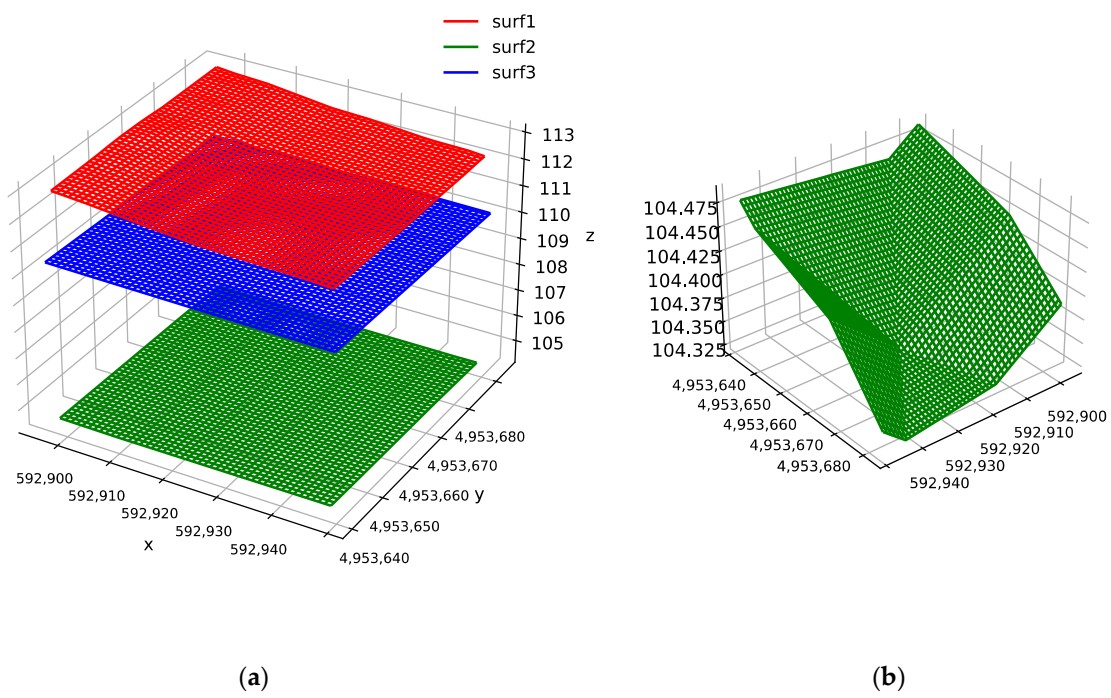

(**a**)　　　　　　　　　　　　　　　　　　　　　　(**b**)

**Figure 5.** The three surfaces with georeferenced coordinates: (**a**) the three surfaces, i.e., the aquifer bottom surface (surf2-green), the upper-level topographic surface (surf1-red), and the groundwater table surface (surf3-blue); (**b**) a zoom of the aquifer bottom surface that highlights a depression in the base aquiclude (see Figure 4).

Initially, we consider a fixed volume of DNAPL (PCE, of density $\rho_n = 1643 \ \text{kg/m}^3$) released in the subsurface that migrates downward in the variably saturated zone. Figure 6 shows the interpolating cubic grid geometry used in this paper. We assume a variably saturated zone to be a parallelepiped of 43.0 m long, from $x = [-8.0, 35.0]$ m (left-hand side), 40.0 m wide from $y = [-20.0, 20.0]$ m (right-hand side), and 13.0 m depth from $z = [102.0, 113.0]$ m. We then consider a spatial resolution of $dx = dy = dz = 1.0$ m and a time step resolution of $dt = 0.1$ s.

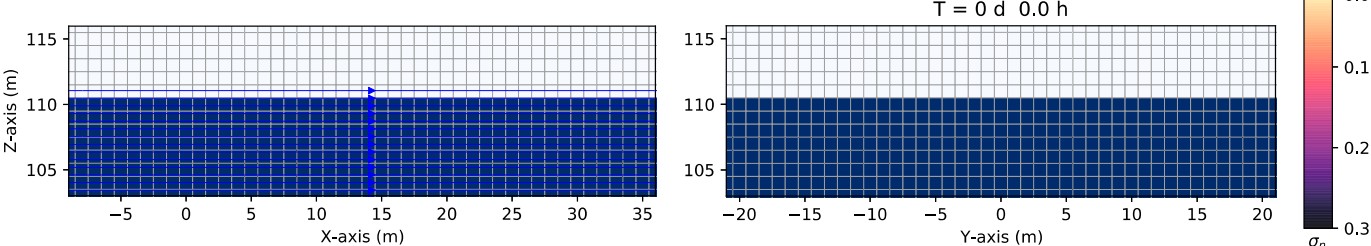

**Figure 6.** Example of the three-dimensional interpolated grid geometry used in the numerical simulation of a three-phase fluid flow (water + DNAPL + air) with a spatial grid resolution of 1.0 m and a grid dimension of 43 m × 40 × 13 m, at the initial time $t = 0$ s. The immiscible contaminant is situated on top of the interpolating cubic grid in the $z - x$ plane view (left-hand side) and the $z - y$ plane view (right-hand side), respectively. The green color in the bottom represents the base aquiclude of the aquifer (Pliocene clay). The green color on the upper side represents the atmosphere. The white color is the unsaturated aquifer. The blue color is the saturated aquifer.

The leakage DNAPL is positioned at $(x, y, z) = (0, 0, 112.0)$ m. The porous medium is composed of an unsaturated dry zone (air-DNAPL), the one depicted in white in Figure 6, and a saturated one, the one depicted in blue, separated by a groundwater table surface (as shown in Figure 5). As previously mentioned, the variably saturated zone is assigned a hydraulic conductivity value equal to $K = 5.0 \times 10^{-5}$ m/s. The impermeable zone below is assigned a value to be $K = 5.0 \times 10^{-12}$ m/s [49]. The green color in the bottom represents the base aquiclude of the aquifer (Pliocene clay). The green color on the upper side represents the atmosphere.

After being released into the environment at atmospheric pressure, the DNAPL migrates downward into the unsaturated zone under the influence of gravity. Figure 7 shows the three-dimensional numerical simulation results of the saturation contours, $\sigma_n = S_n \ \phi$, at different times, for the three-phase immiscible fluid flow (water + DNAPL + air) in a dry unsaturated zone. The saturation contours are viewed in the $z - x$ plane on the left-hand side and in the $z - y$ plane on the right-hand side.

The first panel represents the initial time $t = 0$ s (similar to Figure 6). At the second panel, the DNAPL has reached the groundwater table on the left-hand side and keeps moving vertically since it is denser than the water. Due to an irreducible saturation (see Table 1), part of the DNAPL remains trapped in the porous medium and does not move downward. The rest continues toward the bottom aquifer until it reaches the impermeable surface (see the third panel) and starts to move downstream, to the right-hand side, in agreement with the aquiclude morphology. After 25 days and 14.4 h (the fourth panel), the contaminant moves 5.5 m to the right (in the x direction) and 8.5 m after 57 days and 20.1 h (the last panel). On the right-hand side of this figure, the movement is almost symmetric around the z axes in the y direction.

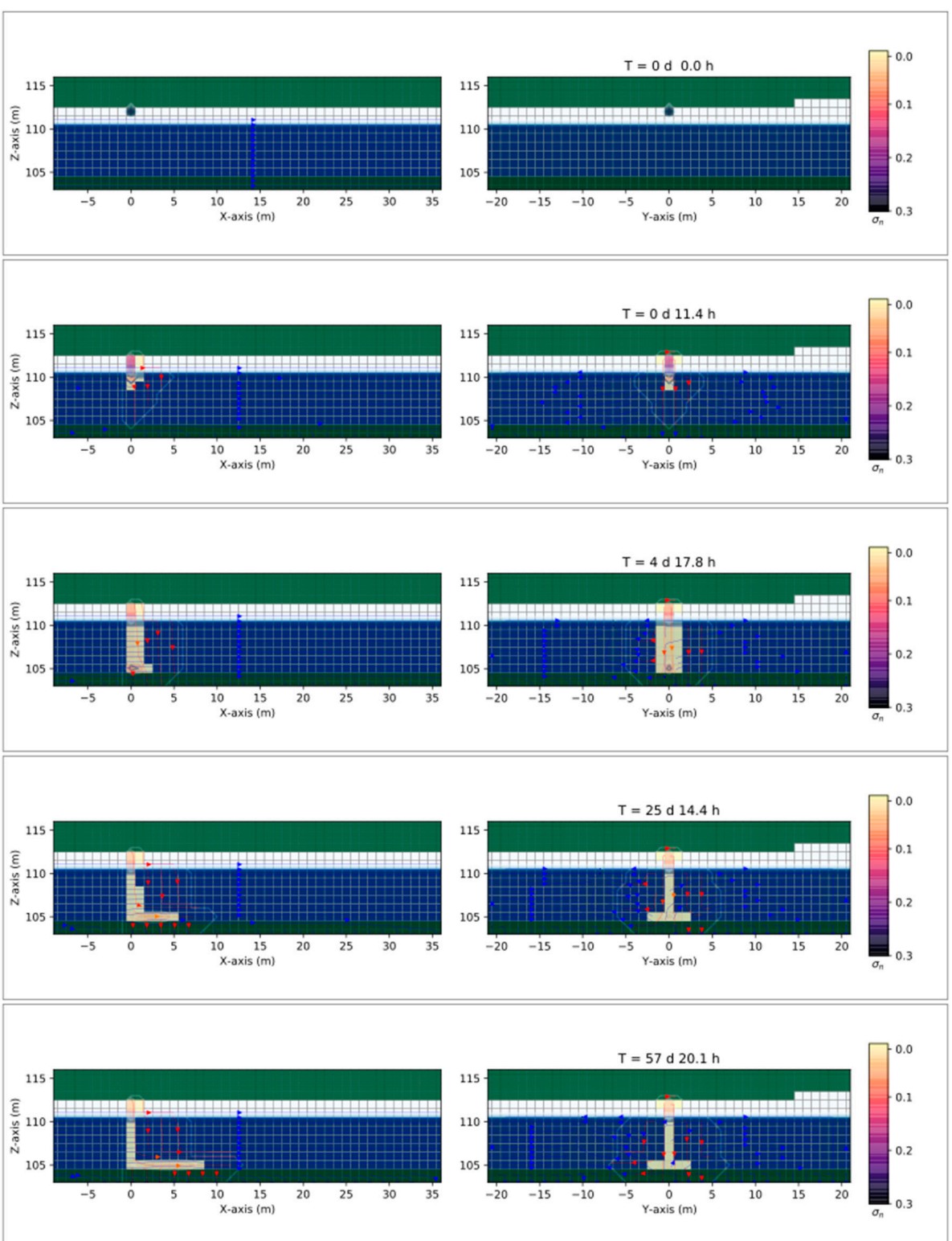

**Figure 7.** Three-dimensional numerical results on the saturation contours ($\sigma_n = S_n \phi$) of the three-phase immiscible fluid flow (water + DNAPL + air) in a dry soil using a spatial grid resolution of 1.0 m, at different times. The DNAPL migrates vertically and then moves downstream in the right direction, in agreement with the aquiclude morphology. The green color in the bottom represents the base aquiclude of the aquifer (Pliocene clay). The green color on the upper side represents the atmosphere. The white color is the unsaturated aquifer. The blue color is the saturated aquifer.

Initially, the mass of DNAPL released into the environment is equivalent to 1643 kg/m$^3$ $\times$ 1 m$^3$ $\times$ 0.37 = 607.9 kg; see Figure 6. The quantity of mass distribution in the entire porous medium after 57 days and a few hours is the product of ($\sigma_n = S_n\phi$) (at this time) multiplied by the DNAPL density, and the volume of the interpolating cubic grid, i.e., 1643 kg/m$^3$ $\times$ (43 $\times$ 40 $\times$ 13) m$^3$ $\times$ 1.65474060822903 $\times$ 10$^{-5}$ = 607.9 kg.

Figure 8 shows the same situation as Figure 7 but includes an extraction pumping after 11 days (the starting time is 1,015,808 s); see the third panel. The well is positioned at the coordinates $(x, y, z) = (3.0, 0.0, 105.0)$ m, specifically at the bottom of the aquifer in the $z$ coordinate. The pumping rate of the extraction is $Q = -0.000119$ m$^3$/s $= -0.119$ L/s, thus in accordance with [51] where the pumping rate was $-0.24$ L/s for a similar aquifer. More in detail,

$$Q_n = \frac{\left(\sigma_{(n,t_f)} - \sigma_{\{n,t_i\}}\right)}{\left(t_f - t_i\right)} \times \text{volume}$$

$$Q_n = \frac{\left(8.129651 \times 10^{-6} - 1.4066608 \times 10^{-5}\right)}{(1,475,379.2 - 1,037,107.2) \text{ s}} \times (40 \times 43 \times 13) \text{ m}^3 = -3.0289490 \times 10^{-7} \text{ m}^3/\text{s}$$

$$Q_w = \frac{\left(\sigma_{(w,t_f)} - \sigma_{\{w,t_i\}}\right)}{\left(t_f - t_i\right)} \times \text{volume}$$

$$Q_w = \frac{(0.25026733699821 - 0.252596564853882)}{(1,475,379.2 - 1,037,107.2) \text{ s}} \times (40 \times 43 \times 13) \text{ m}^3 = -0.00011883381747596284 \text{ m}^3/\text{s}$$

then the total flow rate is the sum of both, the contaminant, and the water flow rate:

$$Q_T = Q_n + Q_w = \begin{array}{l} -3.0289490 \times 10^{-7} - 0.00011883381747596284 \\ = -0.00011913671237778008 \text{ m}^3/\text{s} \end{array}$$

In the fourth panel, the pumping well has extracted part of the free-product DNAPL, as can be noticed on the left-hand side (specifically in the vertical position of the contamination line). In addition, notice the hydraulic head depression due to pumping after 27 days and 11.9 h on both sides. The last panel shows the almost complete recovery of the free-product DNAPL after 62 days and 2.5 h. We calculate the DNAPL left in the entire grid from the numerical simulations. Before starting the pumping, and after 819.0 s, the value of the saturation times, the porosity is $\sigma_n = 1.65474060822899 \times 10^{-5}$. After 62 days and 2.5 h (5,401,804.8 s) $\sigma_n = 5.82892600496648 \times 10^{-6}$. Its ratio is 0.35, which is approximately one-third of the contaminant left. This value agrees with [37] that suggested that DNAPL pools are sometimes large enough to allow effective pumping of the free-product from wells screened at the bottom of the pool. At the same time, free-product pumping usually recovers up to two-thirds of the DNAPL, leaving one-third of DNAPL acting as a long-term pollution source.

The validation of the numerical model against experimental data was performed in a previous publication [26], in which we used the CactusHydro code with the results of a laboratory-scale sandbox with a DNAPL contaminant. The comparison between numerical results and experimental ones shows good agreement and shows that CactusHydro can follow the migration of a plume evolution precisely and can be also used to evaluate the effects and environmental impacts deriving from leaks of DNAPL in variably saturated zones.

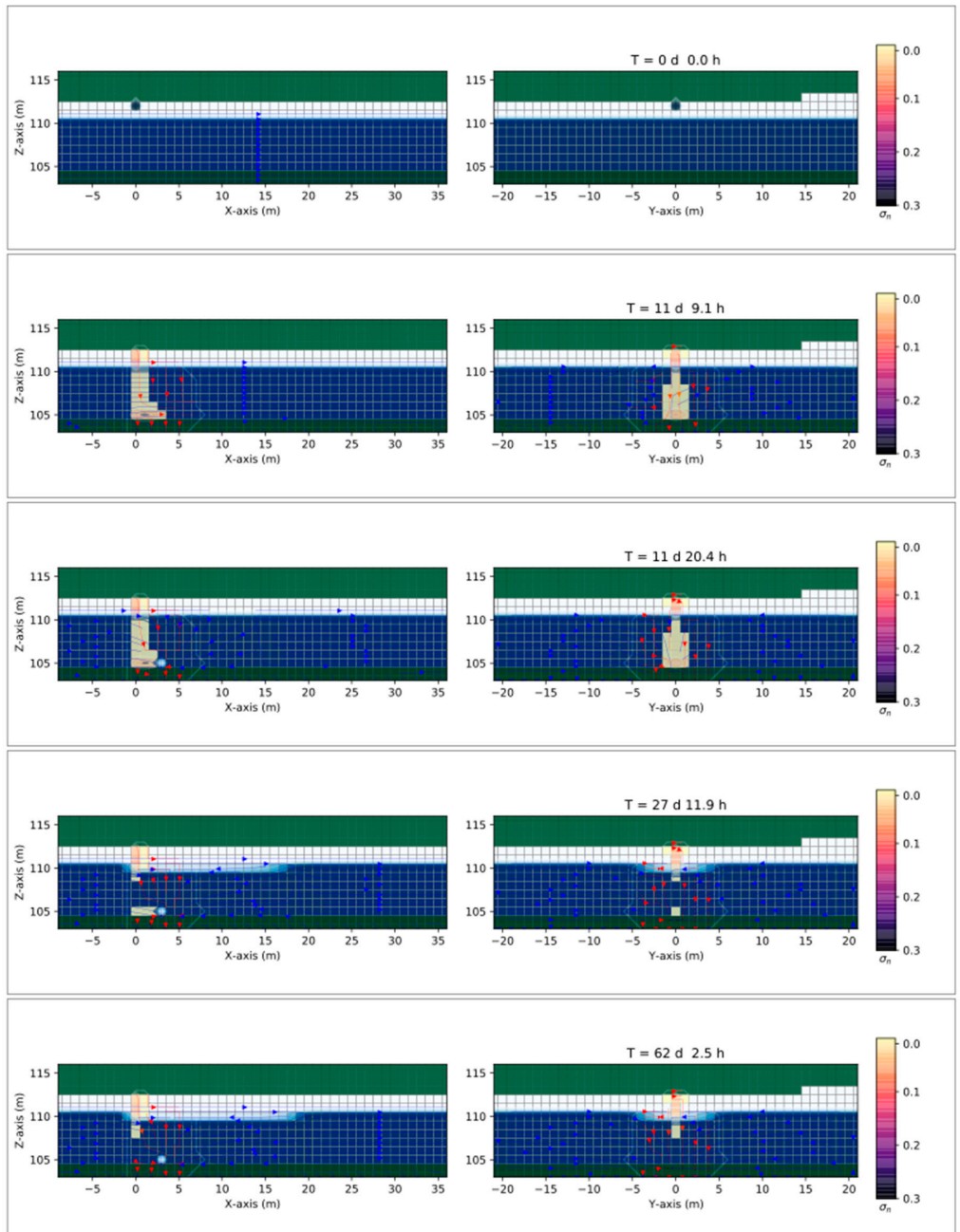

**Figure 8.** Three-dimensional numerical results on the saturation contours ($\sigma_n = S_n\phi$) of the three-phase immiscible fluid flow (water + DNAPL + air) in a dry soil using a spatial grid resolution of 1.0 m, at different times. After 11 days, a pumping well begins to recover the free-product DNAPL (see the third and the fourth panel). The green color in the bottom represents the base aquiclude of the aquifer (Pliocene clay). The green color on the upper side represents the atmosphere. The white color is the unsaturated aquifer. The blue color is the saturated aquifer.

## 4. Discussion and Conclusions

In this work, we presented a three-dimensional numerical modeling investigation on the migration of a free-product DNAPL (PCE) in an alluvial aquifer, the base aquiclude of which was made of Pliocene clay. We followed the numerical results on the saturation contours of the three-phase immiscible fluid flow formed by water, DNAPL, and air in dry soil, at different times, using the CactusHydro [24,25] conservative HRSC method that precisely follows the advective part of the fluid flow and resolves the hyperbolic part of the non-linear governing coupled equations of a three-phase immiscible fluid flow.

Once released in the environment, the DNAPL volume source migrated downward to the bottom of the aquifer while moving together with the flow due to a hydraulic gradient and forming a large enough pool since the DNAPL has slow mobility. We then included the presence of a pumping well screened at the bottom of the pool to recover the contaminant. The results indicate that CactusHydro can precisely follow the contaminant's fate and its recovery using a pumping well. However, the simulation model further demonstrated that the free-product pumping could recover up to two-thirds of the DNAPL, and a significant volume of free-product continues acting as a long-term pollution source, despite the significant contaminant extraction in an emergency scenario. The simulation model allows optimizing the fast design and drilling of pumping wells to stop free-product DNAPL migration towards downgradient and (potentially) deeper aquifer layers. Still, integrative solutions must be planned to remove altogether the long-lasting contamination source that cannot be extracted through wells.

Therefore, interdisciplinary research is in progress to optimize an effective merge between (i) short-term actions to be implemented in the emergency scenario immediately after a DNAPL release (pumping wells) and (ii) medium-term actions to be implemented after removing the mobile free-product DNAPL, such as bioremediation solutions. Concerning the bioremediation perspective, existing data obtained at the study area through biomolecular analyses (community profiling by next generation sequencing; [35]) suggest the existence of an autochthonous bacterial community in the shallow groundwater containing a high percentage of methophiles belonging to several genera, such as *Crenothrix*, *Methylobacter*, *Methylococcus*, and *Methylocella*. These microorganisms are well known to be endowed with methane monooxygenase, suggesting a possible biodegradation pathway of chlorinated solvents through oxidative (co-metabolic) dehalogenation. The vitality of some of these bacteria (e.g., those belonging to the *Methylophilaceae* family) was also verified by enrichment, selection, and isolation procedures.

In terms of practical implications, the approach presented here and the CactusHydro code represent an effective tool to be applied in the environmental risk assessment caused by chlorinated compounds releases, in agreement with the provisions of the Legislative Decree 26 June 2015, n. 105 "Implementation of Directive 2012/18/EU relating to controlling the danger of major accidents connected with dangerous substances" (the so-called Seveso III Directive). Assessing the aquifer vulnerability and analyzing the environmental consequences of an accident are specific components of safety reports to be prepared in accordance with the aforementioned legislation. Therefore, from an application perspective, the simulations made by the CactusHydro code can be used to identify the most effective actions aimed at preventing and/or reducing both the probability and the extent of environmental damage in the case of a significant accident.

**Author Contributions:** Conceptualization, A.F., R.P., A.A. and F.C.; methodology, A.F., R.P. and F.C.; software, A.F.; validation, A.F. and F.C.; formal analysis, A.F., R.P., A.A. and F.C.; investigation, R.P.; resources, A.F., R.P., A.A. and F.C.; data curation, A.F., R.P.; writing—original draft preparation, A.F. and R.P.; writing—review and editing, A.F., R.P., A.A. and F.C.; visualization, A.F. and R.P.; supervision, A.F. and F.C.; project administration, F.C.; funding acquisition, F.C. All authors have read and agreed to the published version of the manuscript.

**Funding:** This research received external funding by Parma Province.

**Data Availability Statement:** The data are contained within the article.

**Acknowledgments:** This work used high-performance computing resources at the University of Parma at (https://www.hpc.unipr.it) accessed on 1 January 2023. This research benefited from the equipment and framework of the COMP-R Initiative, funded by the 'Departments of Excellence' program of the Italian Ministry for University and Research (MUR, 2023–2027). A.F. and F.C. acknowledge financial support from the PNRR MUR project ECS_00000033_ECOSISTER. We also thank two anonymous reviewers for their valuable comments and questions.

**Conflicts of Interest:** The authors declare no conflict of interest.

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
