# Peer review of "Three-Dimensional High-Precision Numerical Simulations of Free-Product DNAPL Extraction in Potential Emergency Scenarios: A Test Study in a PCE-Contaminated Alluvial Aquifer (Parma, Northern Italy)"

_sustainability, doi:10.3390/su15129166_

Round 1
Reviewer 1 Report
The paper by Feo et al describes an high-precision simulations of DNAPL extraction in an aquifer contamination scenario. Chlorinated organic compounds are usually detected at industrial and urban areas and these pollutants are long persistent compounds in the environment, sometimes linked to toxic effects. The article therefore addresses a real and current problem and tries to analyse the benefits, and limitations, of rapid intervention methods in an emergency scenario. From this point of view, the article fills a lack of information, at least in terms of practical application.
The article is well-written, clearly and logically organised and I see no obstacles to its publication. Since it is rather extensive, I believe that the content would benefit somewhat from further, limited information being included in the Introduction (or Methods) in order to enhance its application aspects.
1- first of all, the authors could go into the negative aspects of contamination by Chlorinated organic compounds (rows 35): I realise that this is a very broad field, but I believe that a few lines of description on the best-known effects of acute OOC contamination would help the reader to better understand the necessity and usefulness of the article, which is in any case based on abstract modelling, and to better put the results into reality
2- for the same reasons, the authors should expand the part concerning the wells or catchment points used to analyse the conventration of PCE in the aquifer. Personally, I do not agree with the reference to a figure in another article (R 133 Figure 8 in 33): an article must be self-explanatory and contain all the graphic references needed to understand its contents. It may be permissible under the editorial lines of 'Sustainability' but I would prefer to see Figure 8 in this article. For the same reasons, I feel a brief mention on wells is necessary: what wells are they? I imagine they are institutional monitoring points, or drilled for remediation purposes, or even simple concepts for modelling: either way they need some further explanation for the reader.
.
Reviewer 2 Report
General Comments: The manuscript titled "Three dimensional high-precision numerical simulations of free-product DNAPL extraction in potential emergency scenarios: a test study in a PCE-contaminated alluvial aquifer (Parma, Northern Italy)" presents a numerical model that simulates the migration and extraction of free-product dense non-aqueous phase liquids (DNAPLs) in a potentially hazardous scenario. The study investigates the migration of contaminants in a Parma alluvial aquifer and evaluates the effectiveness of a purpose-designed pumping well in recovering the free-product. The paper provides valuable insights into the behavior of DNAPLs and the potential for their extraction in emergency situations. However, there are several areas that need to be addressed before the manuscript can be considered suitable for publication. The following comments and suggestions are provided to assist the authors in revising their manuscript.
- Clarify the novelty and significance of the proposed numerical approach: The abstract briefly mentions the use of a high-resolution shock-capturing flux conservative method and a new solution for simulating contaminant extraction. However, it would be beneficial to provide more details on the novelty and significance of these approaches. What specific advancements do they offer compared to existing methods? How do they improve the accuracy or efficiency of the simulations? This information will help readers understand the contributions of the study better.
- Provide more details on the numerical model: The abstract states that a numerical model was used to simulate the migration and extraction of DNAPLs. It would be helpful to provide a concise description of the numerical model, including the governing equations, discretization methods, and any assumptions or simplifications made. This information will allow readers to assess the validity and reliability of the simulations.
- Include information on the model validation: The abstract does not mention any validation of the numerical model against experimental or field data. It is essential to include a brief discussion on how the model was validated and its accuracy in reproducing real-world behavior. Providing this information will enhance the credibility of the study and demonstrate the reliability of the simulation results.
- Discuss the implications of the findings: The abstract states that the results indicate the numerical approach's ability to simulate contaminant migration and the recovery of free-product. However, it would be valuable to discuss the implications of these findings in more detail. How do the simulation results contribute to the understanding of DNAPL behavior and extraction in emergency scenarios? What are the practical implications for environmental management and remediation efforts? Elaborating on these points will help readers appreciate the significance of the study's outcomes.
- Provide more specific results: The abstract briefly mentions the recovery of about two-thirds of the free-product, but it lacks specific details regarding the spatial and temporal distribution of the contaminant and the efficiency of the extraction process. Including some key findings or trends in the abstract will make it more informative and encourage readers to delve deeper into the full manuscript.
- Improve the clarity and organization of the abstract: The abstract would benefit from better organization and clarity. Currently, it reads as a collection of separate statements without a clear flow. Consider revising the abstract to provide a logical structure that introduces the problem, outlines the methodology, highlights the main results, and concludes with the significance of the findings.
- Proofread for grammar and language: There are several grammatical errors and awkward sentence structures in the abstract. It is essential to proofread the manuscript carefully to ensure clarity and coherence of the language. This will enhance the overall readability of the abstract.
In summary, the manuscript presents a numerical model for simulating DNAPL migration and extraction in an emergency scenario. While the study offers valuable insights into the behavior of DNAPLs and the effectiveness of a purpose-designed pumping well, there are several areas that need improvement. The authors are encouraged to address the comments and suggestions provided above and revise the manuscript accordingly. With the necessary revisions, the paper has the potential to make a significant contribution to the field and warrant publication.
There are several grammatical errors and awkward sentence structures in the abstract. It is essential to proofread the manuscript carefully to ensure clarity and coherence of the language. This will enhance the overall readability of the abstract.
Round 2
Reviewer 2 Report
The authors have revised the manuscript accordingly.
The quality of the English language has been improved.